# Effects of Jaw-Opening Exercises with/without Pain for Temporomandibular Disorders: A Pilot Randomized Controlled Trial

**DOI:** 10.3390/ijerph192416840

**Published:** 2022-12-15

**Authors:** Shoko Tobe, Hiroyuki Ishiyama, Akira Nishiyama, Keisuke Miyazono, Hiroko Kimura, Kenji Fueki

**Affiliations:** 1Department of Masticatory Function and Health Science, Graduate School of Medical and Dental Sciences, Tokyo Medical and Dental University, Tokyo 113-8549, Japan; 2Department of General Dentistry, Graduate School of Medical and Dental Sciences, Tokyo Medical and Dental University, Tokyo 113-8549, Japan

**Keywords:** temporomandibular disorders, jaw-opening exercise intensity, randomized controlled trial

## Abstract

This study aimed to evaluate the effects of jaw-opening exercises with and without pain on temporomandibular disorders (TMDs), specifically in relation to pain intensity and range-of-mouth opening in patients with TMDs. Participants were randomly assigned to either the jaw-opening exercise with pain (JE w/pain) or the jaw-opening exercise without pain (JE w/o pain) groups, and each exercise was performed for eight weeks. TMDs pain intensity was assessed using a 100-mm visual analog scale (VAS), and the range-of-mouth opening was evaluated at the baseline (T0), 2 weeks (T1), 4 weeks (T2), and 8 weeks (T3). Of the 61 participants, 57 (JE w/pain group, *n* = 30; JE w/o pain group, *n* = 27) were included in the analysis. The range-of-mouth opening and TMDs pain intensity improved from T1 to T3 in both groups. The JE w/pain group showed significant differences at T3 compared to T1 (pain-free unassisted mouth opening, *p* = 0.006; jaw-opening pain, *p* = 0.014; chewing pain, *p* = 0.018). In addition, the JE w/pain group showed significantly greater changes in the maximum unassisted mouth opening at T2 and T3 than the JE w/o pain group (T2, *p* < 0.001; T3, *p* = 0.003). Thus, jaw-opening exercises, until the occurrence of pain, may be effective in patients with TMDs.

## 1. Introduction

Temporomandibular disorders (TMDs) are abnormal functional conditions of the stomatognathic system caused by problems involving the temporomandibular joint (TMJ) and/or masticatory muscles [1]. Among reports of musculoskeletal pain, TMDs are second only to chronic lower back pain, with a prevalence of 5–12% and an annual cost of $4 billion [2]. In the past, TMDs were treated with occlusal therapy; however, around 2000, studies on the natural history of TMDs showed that they improve with time, confirming that TMDs are self-limiting [3]. This led to the thinking that irreversible treatments should not be used in the initial treatment of TMDs, and conservative and reversible treatments have since been recommended [4,5]. Therefore, home care is emphasized in the treatment of TMDs, and exercise therapy and habit guidance are often applied as safe and effective treatment methods [6,7].

Jaw-opening exercises are the standard exercise therapy for TMDs. Several studies have shown that jaw-opening exercises are effective in decreasing muscle and joint pain and restoring TMJ mobility [8,9,10]. However, almost all studies evaluated the effectiveness of jaw-opening exercises, and, to date, no study has examined the effect of jaw-opening exercises on strength and duration [11]. In this study, we used the point-of-pain onset during mouth opening as the reference for jaw-opening exercise intensity, with the aim of determining the effects of jaw-opening exercises with and without pain on TMDs, specifically in relation to pain intensity and the range-of-mouth opening. The null hypothesis was that the improvement in TMDs pain intensity and range-of-mouth opening after jaw-opening exercise with pain is not different from that after jaw-opening exercise without pain.

## 2. Materials and Methods

### 2.1. Trial Design

This was a randomized two-parallel-group clinical trial. The study protocol was reviewed and approved by the Ethics Committee of the Tokyo Medical and Dental University (protocol code: D2018-076, 18 March 2019), and adhered to the tenets of the Declaration of Helsinki. This study was registered with the University Hospital Medical Information Network (UMIN) Registry (UMIN000035260) on 17 December 2018. Participants were recruited in April 2019, and the trial was completed in August 2022.

### 2.2. Participants and Randomization

The candidates for this trial were patients with TMDs diagnosed using the Diagnostic Criteria for Temporomandibular Disorders (DC/TMD) [12] at the TMDs clinic of the Tokyo Medical and Dental University Hospital. The inclusion criteria were as follows: (1) men and women aged ≥ 18 years; (2) patients with pain-related TMDs, including myalgia (masseter and/or temporalis), arthralgia, or both as the TMDs subtype; (3) TMDs pain for at least one month, with no change or worsening of symptoms; (4) pain-free unassisted mouth opening < 40 mm; and (5) an average pain intensity during jaw opening ≥ 31 mm on a 100-mm Visual Analog Scale (VAS) [13]. The exclusion criteria were as follows: (1) an anterior disc displacement without reduction or osteoarthritis as the TMDs subtype; (2) a history of surgery on the TMJ or masticatory muscles; (3) patients with neurological and psychiatric disorders; (4) patients who regularly use anxiolytic or analgesic drugs that affect pain; and (5) those who had previously undergone jaw-opening exercise under the guidance of a dentist for their current pain. Patients who underwent massage or splint therapy were not excluded. All participants provided written informed consent.

After enrollment, participants were randomly assigned to either the jaw-opening exercise with pain (JE w/pain) group or the jaw-opening exercise without pain (JE w/o pain) group. We used a block size of 4 to ensure equal allocation according to sex and age between the two groups, and randomization was performed by an independent researcher using the envelope method. The participants were divided into three groups according to age, based on the data on TMJ abnormalities from the Dental Disease Survey 2016 [14]: 18–34 years, 35–54 years, and ≥55 years. In addition, in all three age groups, participants were further grouped according to sex, for a total of six sets of envelopes. These envelopes were prepared by an independent researcher. The dentists who instructed on the jaw-opening exercises and who assessed the outcomes were different researchers, and information on the group assignment was blinded to the examiners. Participants were informed at the time of enrollment that both jaw-opening exercises were effective in treating TMDs, and were blinded to the exercises that were actually assigned.

### 2.3. Intervention

In the JE w/pain group, as jaw exercises, participants were instructed to manually open their mouths at home according to the following protocol (Figure 1) [15,16]. Three fingers of the dominant hand, excluding the thumb and little finger, were to be placed on the incisal edges of the mandibular incisors, and the thumb of the opposite hand was to be placed on the incisal edges of the maxillary incisors. Participants were to open their mouths with their hands and keep them open when they started to perceive pain in the TMJs and/or masticatory muscles (masseter and/or temporalis). The open position was to be maintained for 10 s. Participants performed this stretching five times as a single set after each meal and before bedtime; a total of four sets were performed each day for eight weeks.

In the JE w/o pain group, participants were instructed to perform the jaw-opening exercise without using their hands, by themselves at home, according to the following protocol (Figure 2). Participants were to open their mouths as wide as possible without any pain in their masticatory muscles (masseter and/or temporalis) or TMJs. The open position was to be maintained for 10 s. The participants performed this stretching five times as a single set, after each meal and before bedtime; a total of four sets were performed each day for eight weeks.

The appropriateness of the exercises performed by each participant was confirmed in this study. If inappropriate, we instructed the participants on the exercise method at each follow-up.

### 2.4. Outcome Measures

The outcomes at the baseline (T0), 2 weeks (T1), 4 weeks (T2), and 8 weeks (T3) of jaw-opening exercise were evaluated. TMDs pain intensity (during jaw opening and chewing) and range-of-mouth opening (pain-free unassisted mouth opening and maximum mouth opening) were evaluated as primary outcomes. Difficulty in performing the exercise and the implementation of the exercise were evaluated as secondary outcomes. TMDs pain intensity was measured using a 100-mm Visual Analog Scale (VAS), with “no pain” at the left end and “intolerable pain” at the right end. Participants were instructed to mark a vertical line on the scale at the point indicating the average TMDs-related pain they had experienced during jaw opening and chewing in the past week. The VAS score was defined as the distance from the left to the vertical line. The range-of-mouth opening was measured using calipers in accordance with DC/TMD [12]; the overbite was then added to the measured value to calculate the actual amount of movement. Additionally, the pain-free unassisted mouth opening and maximum unassisted mouth opening were assessed. Pain-free unassisted mouth opening was defined as the maximum distance that the participants could open their mouths without experiencing pain. Maximum unassisted mouth opening was defined as the maximum distance that the participants could open their mouths even if they felt pain.

The difficulty in performing the exercise and its implementation were evaluated using a self-administered questionnaire. The questions (and possible responses) were: ‘How difficult was it for you to do the jaw-opening exercises?’ (1, very difficult; 2, somewhat difficult; 3, cannot say either way; 4, not very difficult; and 5, not at all difficult); ‘How well did you perform the jaw-opening exercise?’ (1, could not do it at all; 2, not much; 3, cannot say either way; 4, somewhat well; and 5, very well).

### 2.5. Sample Size Estimation

The sample size was calculated using the G*Power software. There were no data from previous studies on the change in VAS scores for TMDs pain intensity with different intensities of jaw-opening exercises. Therefore, we assumed the effect size on the change in the VAS score to be medium (0.5). Accordingly, 70 participants per group, including an assumed dropout rate of 10%, were required to achieve 80% power with an alpha level of 5%.

### 2.6. Statistical Analysis

In this study, missing values were handled using simple imputation (personal worst score) [17]. Participants without any post-baseline data were defined as dropouts and excluded from the analysis. The pain sensitivity of TMDs varied significantly among individuals [18]. Therefore, in this study, the changes in TMDs pain intensity and range-of-mouth opening from T0 were calculated at each follow-up. For statistical analysis, a Student’s *t*-test and chi-square test were used for T0 comparisons between groups. One-way analysis of variance (ANOVA) with repeated measures was used to evaluate the time–progress changes in the range-of-mouth opening and TMDs pain intensity within the groups. A Student’s *t*-test was used for comparisons of the range-of-mouth opening and TMDs pain intensity between groups at each follow-up. With regard to the difficulty in performing the exercise and implementation of the exercise at each follow-up, a chi-square test was used to compare the distribution of questionnaire responses between the groups. For multiple comparisons, *p*-values were corrected using the Bonferroni correction. The effect size was calculated (Cohen’s d) to evaluate the clinical relevance of the magnitude of the differences (0.2, small; 0.5, medium; and 0.8, large) between the groups and not only the statistical significance. All statistical analyses were performed using the SPSS software version 21.0 (IBM, Inc., Armonk, NY, USA). Statistical significance was set at *p* < 0.05.

## 3. Results

### 3.1. Participants

Sixty-six patients with TMDs were recruited. Five patients who did not meet the inclusion criteria for this study were excluded (three were too busy to visit the hospital and two refused to participate in the study). Thus, 61 patients (49 women and 12 men, from 18–90 years old, with a mean age of 50.1) were enrolled in the study. These patients were assigned to two groups: 31 in the in the JE w/pain group and 30 participants in the JE w/o pain group. Figure 3 shows the flow diagram of the trial and participant assessment. A total of 57 participants (46 women and 11 men, from 18–90 years old, with a mean age of 51.4 years) were finally included in the analysis after four dropped out. In the JE w/pain group, one participant dropped out because of a refusal to visit due to fear of the coronavirus disease (COVID-19). In the JE w/o pain group, three participants dropped out (one because of a refusal to visit due to fear of COVID-19, one contracted COVID-19, and the third due to difficulty attending the clinic due to a job search).

The baseline characteristics of the participants in the two groups (*n* = 61, including dropouts) are shown in Table 1. The prevalence of myalgia, arthralgia, and myalgia with arthralgia in the TMDs subtype was significantly different between the two groups (*p* = 0.005). There were no significant differences in the other parameters, and the characteristics of the two groups were comparable (*p* > 0.05).

### 3.2. Assessment of Primary Outcomes

The change in TMDs pain intensity from T0 to each follow-up in the two groups is shown in Table 2. In the JE w/pain group, significant changes were observed in both jaw-opening and chewing pain at some points during the 8-week period (jaw-opening pain, *p* < 0.001; chewing pain, *p* = 0.003). In multiple comparisons, jaw-opening and chewing pain in the JE w/pain group at T3 were significantly higher than those at T1 (jaw-opening pain, *p* = 0.014; chewing pain, *p* = 0.018). In contrast, jaw-opening and chewing pain were not significantly different in the JE w/o pain group during the 8-week period (*p* > 0.05). There was no significant difference in the change in jaw-opening or chewing pain at each follow-up (*p* > 0.05).

The amount of change from T0 to each follow-up in the two types of unassisted mouth-opening ranges is shown in Table 3. In the JE w/pain group, repeated one-way ANOVA revealed a significant change in pain-free unassisted mouth opening after eight weeks (*p* = 0.004). In multiple comparisons, the change in pain-free unassisted mouth opening at T3 was significantly higher than that at T1 in the JE w/pain group (*p* = 0.006). In contrast, in the JE w/o pain group, no significant change was observed in pain-free unassisted mouth opening (*p* > 0.05). There was no significant difference between the two groups in the change in pain-free unassisted mouth opening at each follow-up (*p* > 0.05). In both groups, no significant change was observed in the maximum unassisted mouth opening at any evaluation during the 8-week period (JE w/pain group, *p* = 0.12; JE w/o pain group, *p* = 0.49). However, the change in maximum unassisted mouth opening at T2 and T3 in the JE w/pain group was significantly higher than that in the JE w/o pain group (T2, *p* < 0.001; T3, *p* = 0.003).

### 3.3. Assessment of Secondary Outcomes

Figure 4 is a band graph showing the percentage of the responses to the question, ‘How difficult was it for you to do the jaw-opening exercise?’. There was no significant difference between the two groups in the percentage of the responses to the question on the difficulty of the jaw-opening exercise at any evaluation (*p* > 0.05). In the JE w/o pain group, no participant reported the jaw-opening exercise as ‘Very difficult’ from T1 to T3, whereas in the JE w/pain group, 10%, 6.7%, and 6.7% of participants reported that the jaw-opening exercise was ‘Very difficult’ at T1, T2, and T3, respectively. The percentage of participants who reported that the jaw-opening exercise was ‘Somewhat difficult’ was 14.8%, 22.2%, and 18.5%, and 33.3%, 46.7%, and 40.0% at T1, T2, and T3 in the JE w/o pain and JE w/pain groups, respectively.

The proportion of participants’ responses regarding the implementation of the exercise at each follow-up is shown in Figure 5. There was no significant difference in the response distributions between the two groups at any follow-up (*p* > 0.05). ‘Somewhat well’ or ‘Very well’ represented >90% of the total responses to the question ‘How well did you perform the jaw-opening exercise?’ in both groups. In addition, only 3.3% of the participants in the JE w/pain group reported ‘Could not do it at all’ from T1 to T3. In the JE w/o pain group, 3.7% of participants reported that they “Could not do it at all” at T1, but not at T2 and T3.

## 4. Discussion

To the best of our knowledge, this is the first study to determine the effects of jaw-opening exercises with and without pain on TMDs pain intensity and range-of-mouth opening. In the treatment of TMDs, it is clinically important to verify whether patients forcefully open their mouth until they experience pain. This is because dentists use this information as a recommendation when explaining the intensity of jaw exercises to patients. Stretching is performed to improve joint mobility and muscle flexibility in the whole body [19,20], and is divided into static and dynamic stretching [21]. The jaw-opening exercise in this study was considered static stretching because the TMJs and masticatory muscles were slowly lengthened and maintained in a stretched state without recoil or bounce [22]. Participants in this study were patients with TMDs who had been aware of their pain for at least one month and whose symptoms remained unchanged. Collins et al. reported that when pain was assessed using a 100-mm VAS, those with a pain rating ≥ 31 mm experienced moderate pain, whereas those with a pain rating ≥ 54 mm experienced severe pain [13]. In this study, all participants had a pain rating ≥ 31 mm during jaw opening; thus, they had moderate or severe pain.

In this study, the maximum unassisted mouth opening significantly increased when the participants opened their mouths until they experienced pain. Takeuchi et al. compared the effects of three different stretching-strength levels for 20 s on the knee joint [23]. The range-of-joint movement increased more when stretching was performed with stronger force, and the stiffness of the muscle-tendon complex decreased. Furthermore, they reported that low-force stretching did not improve muscle flexibility. This is consistent with our results in the present study, in which the JE w/pain group showed a greater increase in the mouth-opening range than the JE w/o pain group. The range of TMJ motion and flexibility of the masticatory muscles are more likely to be improved by training the patient to open the jaw more forcefully until pain occurs. An increase in joint range of motion improves flexibility, blood flow, and nutrient supply to the muscles [24]. For joints, this reduces inflammation, rescues synovial cells from dysfunction, and protects the cartilage [25]. On the other hand, a decrease in the range-of-mouth opening causes not only persistent TMDs pain due to low blood flow to the muscle [26], but also other inconveniences, such as difficulty in eating large foods, interference with intraoral examination and treatment during dental care [27,28], and increased difficulty in securing the airway during respiratory distress [29]. It is important to increase patients’ range-of-mouth openings because the elimination of these inconveniences can significantly improve their quality of life. [30] Therefore, it is clinically significant that the present study revealed that the range-of-mouth opening increased more when patients opened their mouths forcefully to the point of pain.

In this study, the JE w/pain group showed a greater improvement in both the range-of-mouth opening and TMDs pain, and the maximum unassisted mouth opening differed significantly between the groups. However, there was no significant difference between the groups in pain reduction. This may be attributed to the large individual differences in pain perception. At the baseline, the standard deviations of jaw-opening pain and chewing pain were large in both groups. Furthermore, the small sample size may have affected our results.

If the intensity of the jaw-opening exercise is increased until pain occurs, there is concern not only about the effect of the treatment, but also about a decrease in adherence to the treatment. Therefore, in the present study, two types of jaw-opening exercises were conducted, and the degree of exercise difficulty and implementation were evaluated. The percentage of participants who felt that the exercise was difficult tended to be higher in the JE w/pain group than in the JE w/o pain group at all evaluation periods. However, no participant dropped out of the study because of difficulty in performing the exercise. There was no significant difference between the groups in the degree of exercise implementation at any evaluation period, and >90% of the participants answered that the exercise was performed ‘Somewhat well’ or ‘Very well’. Therefore, jaw-exercise intensity was thought to have little effect on treatment adherence. However, careful guidance and observation are necessary when performing jaw-opening exercises, as pain may worsen in daily life.

This study had several limitations. First, the intervention of this study depended on the participant’s self-care, and then the subjective sensation of pain was evaluated as a primary outcome. Furthermore, bias toward the researcher who instructed on the jaw exercise was unavoidable. These could have affected the reliability and validity of the results. Therefore, the confirmation of adherence to the jaw exercise was evaluated by participant self-assessment alone using a questionnaire, but a diary or application should have been added to the survey to make it more reliable [31]. Second, in this study, the strength of the jaw-opening exercise used the point-of-pain onset during mouth opening as a reference point. However, because the force required during the jaw-opening exercise was not objectively standardized, it was unclear whether the participants in the JE w/pain group actually opened their mouths with more force during the jaw-opening exercise than those in the JE w/o pain group. Moreover, the intensity of the jaw-opening exercises performed by the participants was unclear because it was not possible to check whether the JE w/pain group opened to the extent that it became painful, as instructed, nor whether the JE w/o pain group opened to the maximum extent without it being painful. However, we checked if the jaw-opening exercise performed by the participants was appropriate at each follow-up, and no one performed the exercise inappropriately. Third, at the baseline, there were significant differences in the TMDs subtype between the two groups. Group comparisons of TMDs pain intensity and range-of-mouth opening according to TMDs subtype showed the same tendency as those without, considering the TMDs subtype as a confounder. Therefore, the TMDs subtype had a limited influence on the study results. To evaluate the effect of the jaw-opening exercise on the TMDs pain disturbance of mouth opening, we excluded any disc displacement without reduction in which the sliding motion of the condyle was obstructed. The DC/TMD used in this study can diagnose disc displacement without reduction with a sensitivity of 0.80 and a specificity of 0.97 on clinical examination without imaging examination [12]. Therefore, disc displacement without reduction was mostly excluded at the time of recruitment. However, since magnetic resonance imaging was not performed, it is possible that some participants may have had disc displacement without reduction. Finally, the sample size of the present study was small. The number of patients who came to the hospital was low owing to the COVID-19 pandemic, and few met the eligibility criteria. Therefore, the number of participants in this study was much smaller than the sample size originally calculated. We believe that an analysis with a larger sample size is required.

The following are issues for future research. Diagnosis using magnetic resonance imaging should be added to the eligibility criteria to characterize the participants more uniformly. In this study, stratified randomization was performed for sex and age, but we would also include the TMDs subtype to more reliably assign the two groups equally. For surveys of home-based self-care adherence, using an app to ensure that the self-care adherence is recorded without delay should provide more accurate confirmation that the intervention was properly implemented. Furthermore, we think that by collecting enough samples and then analyzing them, we would be able to produce more reliable research results. In addition to the presence or absence of pain in the jaw-opening exercises, other variables related to jaw exercises, such as the time necessary to maintain mouth opening and jaw exercise other than slow stretching, should also be investigated to provide more specific exercise therapy instructions to the patients.

## 5. Conclusions

The present study confirmed that TMDs pain intensity reduced over time after jaw-opening exercises with and without pain, although there was no difference between the groups. However, the maximum range of unassisted mouth opening improved to a greater degree by performing the jaw-opening exercise with pain. The impact of the jaw-opening exercise with pain on treatment adherence was small, and no one discontinued the study due to pain. Consequently, when instructing patients with pain-related TMDs in jaw-opening exercises, it may be more effective to open their mouths until pain occurs. Further studies with larger sample sizes are needed to produce more reliable results.

## Figures and Tables

**Figure 1 ijerph-19-16840-f001:**
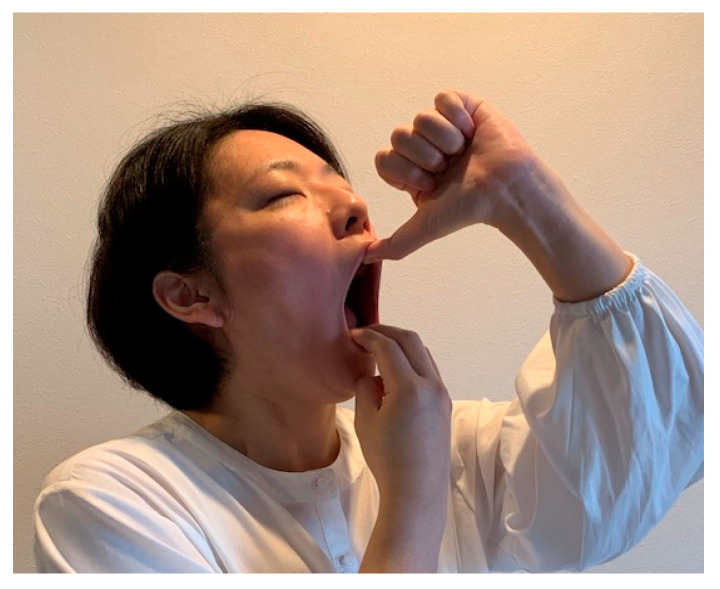
Jaw-opening exercise with pain (JE w/pain group): using their own hands, the participants performed the jaw-opening exercise passively to the point of pain in the TMJs and/or masticatory muscles (masseter and/or temporalis). The opened position was maintained for 10 s.

**Figure 2 ijerph-19-16840-f002:**
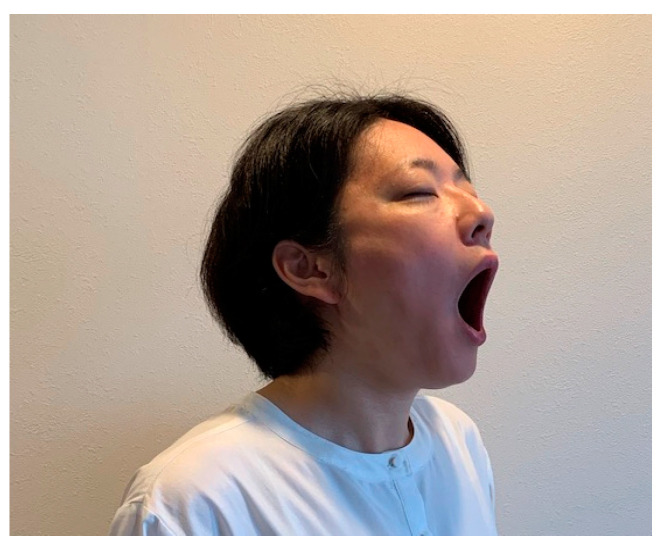
Jaw-opening exercise without pain group (JE w/o pain group): the participants performed the jaw-opening exercise without using their hands, by themselves. The opened position was maintained for 10 s.

**Figure 3 ijerph-19-16840-f003:**
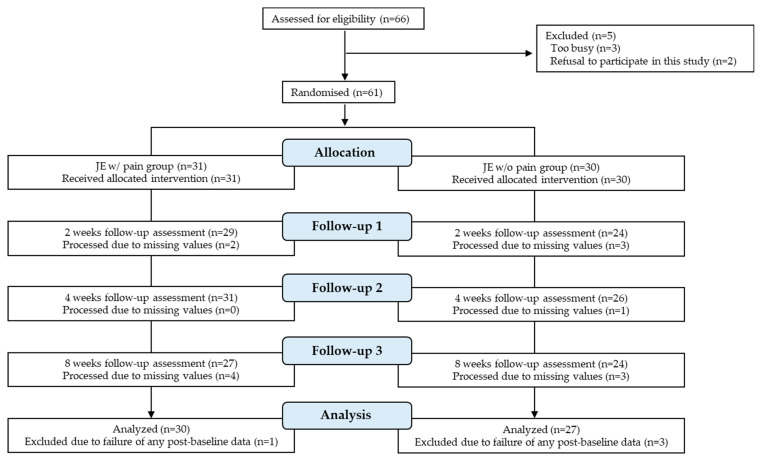
CONSORT flow diagram for our randomized controlled trial.

**Figure 4 ijerph-19-16840-f004:**
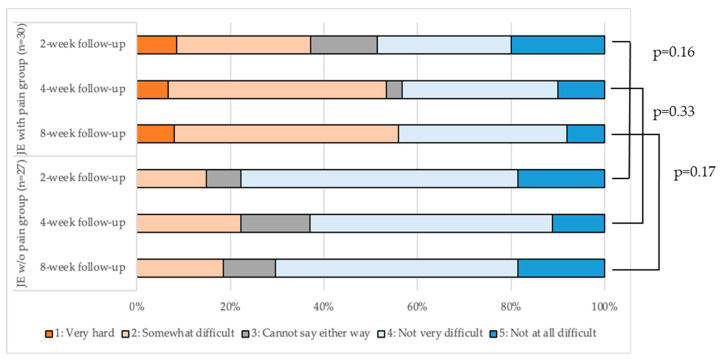
Evaluation of the difficulty of jaw-opening exercise in the two groups.

**Figure 5 ijerph-19-16840-f005:**
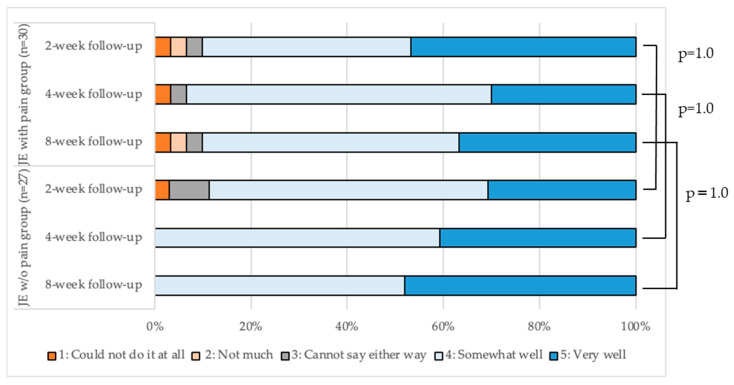
Evaluation of the implementation of the jaw-opening exercise in the two groups.

**Table 1 ijerph-19-16840-t001:** Comparison of baseline demographic and outcome variables between the two groups.

Variables	Total (*n* = 61)	JE w/Pain Group (*n* = 31)	JE w/o Pain Group (*n* = 30)	*p*-Value for Group Difference
Age (years) ^a^	50.1 (19.3)	48.7 (19.8)	51.6 (19.1)	0.56
Sex ^b^				
Male	12 (19.7)	7 (22.6)	5 (16.7)	0.75
Female	49 (80.3)	24 (77.4)	25 (83.3)	
Disease duration (months) ^a^	17.6 (44.5)	15.7 (45.0)	19.5 (44.6)	0.74
TMDs subtype according to the DC/TMD ^b^				
Myalgia	28 (45.9)	12 (38.7)	16 (53.3)	0.005 *
Arthralgia	15 (24.6)	13 (41.9)	2 (6.7)	
Myalgia + Arthralgia	18 (29.5)	6 (19.4)	12 (40)	
Pain intensity (VAS, mm) ^a^				
Jaw-opening pain	46.9 (11.8)	45.3 (12.3)	48.6 (11.2)	0.27
Chewing pain	33.2 (22.2)	33.5 (22.5)	32.8 (22.2)	0.91
Active mouth opening (mm) ^a^				
Without pain	30.6 (6.3)	29.9 (6.6)	31.3 (5.9)	0.41
Maximum mouth opening	41.4 (7.3)	40.8 (8.2)	42.0 (6.3)	0.53

TMDs: Temporomandibular Disorders, VAS: Visual Analog Scale, DC/TMD: Diagnostic Criteria for Temporomandibular Disorders. ^a^ Mean (standard deviation). ^b^ Number of participants (percentage). * *p* < 0.05.

**Table 2 ijerph-19-16840-t002:** Changes in temporomandibular disorders (TMDs) pain intensity in the two groups.

Amount of Change from Baseline in VAS Score (mm)		2-Week Follow-Up	4-Week Follow-Up		8-Week Follow-Up		*p*-Value ^b^
		Mean (SD)	Mean (SD)	*p*-Value ^a^	Mean (SD)	*p*-Value ^a^	
Jaw-opening pain	JE w/pain group (*n* = 30)	−12.9 (18.1)	−23.7 (19.2)	0.01 *	−24.8 (22.8)	0.014 *	<0.001 *
	JE w/o pain group (*n* = 27)	−17.0 (15.7)	−17.5 (18.2)	1.0	−26.9 (20.0)	0.4	0.064
	*p*-value ^c^	1.0	0.45		1.0		
	Effect size (d)	0.24	0.33		0.10		
Chewing pain	JE w/pain group (*n* = 30)	−4.5 (14.3)	−13.8 (15.7)	0.002 *	−15.9 (23.7)	0.018 *	0.003 *
	JE w/o pain group (*n* = 27)	−8.2 (17.0)	−8.0 (18.8)	1.0	−14.3 (24.5)	0.44	0.17
	*p*-value ^c^	1.0	0.45		1.0		
	Effect size (d)	0.24	0.34		0.07		

^a^ Multiple comparisons among the 2-, 4-, and 8-week follow-ups. ^b^ Overall assessments based on repeated one-way ANOVA. ^c^ Comparison between the two groups based on Student’s *t*-test with Bonferroni correction. VAS: Visual Analog Scale. * *p* < 0.05.

**Table 3 ijerph-19-16840-t003:** Changes in mouth-opening range in the two groups.

Amount of Change from Baseline in Mouth-Opening Range (mm)		2-Week Follow-Up	4-Week Follow-Up		8-Week Follow-Up		*p*-Value ^b^
		Mean (SD)	Mean (SD)	*p*-Value ^a^	Mean (SD)	*p*-Value ^a^	
Pain-free unassisted mouth opening	JE w/pain group (*n* = 30)	5.8 (7.5)	10.7 (7.4)	0.052	11.3 (8.5)	0.006 *	0.004 *
	JE w/o pain group (*n* = 27)	5.7 (6.3)	6.4 (7.1)	1.0	8.2 (6.8)	1.0	0.37
	*p*-value ^c^	1.0	0.11		0.48		
	Effect size (d)	0.01	0.59		0.40		
Maximum unassisted mouth opening	JE w/pain group (*n* = 30)	5.7 (6.3)	6.4 (7.1)	1.0	8.2 (6.8)	0.26	0.12
	JE w/o pain group (*n* = 27)	0.1 (7.4)	1.0 (6.2)	1.0	2.2 (4.9)	1.0	0.49
	*p*-value ^c^	0.20	<0.001 *		0.003 *		
	Effect size (d)	0.82	0.81		1.0		

^a^ Multiple comparisons among the 2-, 4-, and 8-week follow-ups. ^b^ Overall assessments based on repeated one-way ANOVA. ^c^ Comparison between the two groups based on Student’s *t*-test with Bonferroni correction. * *p* < 0.05.

## Data Availability

The data presented in this study are available upon request from the corresponding author.

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
