# Peer review of "Effects of Jaw-Opening Exercises with/without Pain for Temporomandibular Disorders: A Pilot Randomized Controlled Trial"

_ijerph, 2022, doi:10.3390/ijerph192416840_

Round 1
Reviewer 1 Report
Thank you for your contribution of the journal. Your study is clinically so important that jaw-opening exercise with the amount of mouth open until pain occurs is more effective for maximum unassisted mouth opening while both jaw-opening exercises had effective to reduce jaw-opening and chewing pain. However, I have some concerns to publish the manuscript. Please revise your manuscript with my comments to the following.
1. Title: In this study, the patient was instructed the jaw-opening exercise with a painfully large amount of opening mouth and the jaw-opening exercise to open the mouth to a non-painful extent. So, the expression “with/without” seems to have a slightly different nuance. Also, "jaw-opening exercise with high force and pain" and "jaw-opening exercise with low force and without pain" (P2L47) will be misleading because it did not set the force of the jaw-opening exercise in this case. Please mention in the method the degree of opening during opening exercises with pain (opening to the point where pain begins to occur? Unassisted maximum opening volume).
2. (P2L80) Have you tried to remove bias from the doctor who instructed the opening training?
3. (Fig.3) The number of people is mentioned in Exclusion of Analysis, but the reason is not mentioned. Please add it.
4. (Table 1) It is supposed to indicate the Number of patients and Percentage, but I think it does not indicate this for disease duration and opening amount. Please correct it so that we can understand what it represents. Also, please add the unit of measure for each.
5. The order of the descriptions of "Opening amount" and "Pain" is not consistent throughout the entire document (In Table 1, the order is Opening amount → Pain, and in Table 2 & 3, the order is Pain → Opening amount).
6. (3.3 Assessment of secondary outcomes) How did you binarize against the answers to the questions in order to perform a chi-square test? Please add this to the statistical analysis.
7. (P8L256) Please correct 100-mm visual analog scale to 100-mm VAS.
8. This study is a pilot RCT, but if the number of subjects is only small, it will be an interim report. To make it a pilot study, please describe a specific future plan based on the results of this study and what kind of analysis will be added in future studies.
Author Response
Point 1: Title: In this study, the patient was instructed the jaw-opening exercise with a painfully large amount of opening mouth and the jaw-opening exercise to open the mouth to a non-painful extent. So the expression “with/without” seems to have a slightly different nuance. Also, "jaw-opening exercise with high force and pain" and "jaw-opening exercise with low force and without pain" (P2L47) will be misleading because it did not set the force of the jaw-opening exercise in this case. Please mention in the method the degree of opening during opening exercises with pain (opening to the point where pain begins to occur? Unassisted maximum opening volume).
Response 1: The participants in this study were TMDs patients with jaw-opening pain, and We changed the manuscript regarding the degree of force of the jaw-opening exercise in the Jaw w/ pain group as follows (line:90-92).
“Participants opened their mouth with their hands and keep them open when they started to perceive pain in the TMJs and/or masticatory muscles (masseter and/or temporalis)”.
Moreover, we added the manuscript to the limitation of the discussion as follows (line:310-320).
“Second, in this study, the strength of the jaw-opening exercise was used the point of pain onset during mouth opening as a reference point. However, because the force required during jaw-opening exercise was not objectively standardized, it was imprecise whether the participants in the JE w/ pain group actually opened their mouths with more force during jaw-opening exercise than the JE w/o pain group. Moreover, the intensity of the Jaw-opening exercises performed by the participants was unclear because it was not possible to check whether the JE w/ pain group opened to the extent that it was painful as instructed, or whether the JE w/o group opened to the maximum extent that it was not painful. However, we checked if the jaw-opening exercise performed by the participants was appropriate at each follow-up, and no one performed the exercise inappropriately.”
Point 2: (P2L80) Have you tried to remove bias from the doctor who instructed the opening training?
Response 2: We agree with the reviewer. Bias toward researchers who instructed the jaw-opening exercise was unavoidable. We added explanatory text to the discussion as follows (line:306-307).
Point 3: (Fig.3) The number of people is mentioned in Exclusion of Analysis, but the reason is not mentioned. Please add it.
Response 3: We added the reason for Exclusion in Fig.3.
Point 4: (Table 1) It is supposed to indicate the Number of patients and Percentage, but I think it does not indicate this for disease duration and opening amount. Please correct it so that we can understand what it represents. Also, please add the unit of measure for each.
Response 4: We agree with the reviewer. We corrected Table 1.
Point 5: The order of the descriptions of "Opening amount" and "Pain" is not consistent throughout the entire document (In Table 1, the order is Opening amount→Pain, and in Table 2 & 3, the order is Pain→Opening amount).
Response 5: We unified the order of the range of mouth opening and the pain by switching them in Table 1.
Point 6: (3.3 Assessment of secondary outcomes) How did you binarize against the answers to the questions in order to perform a chi-square test? Please add this to the statistical analysis.
Response 6: In this study, the distribution of responses in the questionnaire was not binarized. A cross-tabulation table was created to evaluate the distribution of responses on the 5-point scales between the two groups, and a chi-square test was conducted.
Point 7: (P8L256) Please correct 100-mm visual analog scale to 100-mm VAS.
Response 7: We corrected 100-mm visual analog scale to 100-mm VAS.
Point 8: This study is a pilot RCT, but if the number of subjects is only small, it will be an interim report. To make it a pilot study, please describe a specific future plan based on the results of this study and what kind of analysis will be added in future studies.
Response 8: We added a future research issue to the Discussion section (line:337-348).
“The following are issues for the future research. Diagnosis using magnetic resonance imaging will be added to the eligibility criteria to characterize the participants more uniformly. In this study, stratified randomization was performed for sex and age, but we will also add TMDs subtype to assign the two groups equally more reliably. For surveys of home-based self-care adherence, preferably using an app to ensure that it was recorded without delay, should provide more accurate confirmation that the intervention was properly implemented. Furthermore, we think that by collecting enough samples and then analyzing them, we will be able to produce more reliable research results. In addition to the presence or absence of pain in the jaw-opening exercises, other variables related to jaw exercises, such as time to maintain mouth opening and jaw exercise other than slow stretching, should also be investigated to provide more specific exercise therapy instructions to the patients.”

Reviewer 2 Report
Thank you for the opportunity to review this article.
It is a very interesting study, designed to check effectiveness of physiotherapy programs.
The design is carefully made, although the major flaw - not mentioned by the authors as a weakness - is it depends on self action by the patients and involves subjective measurement as pain. It should be clearly explained and stated in discussion as it is a has a major influence on the validity and impact of the results.
The results section should be slightly re-written - the presentation is not always clear and easy for understanding.
Finally, would appreciate clearer conclusions with advice for future research.
Author Response
Point 1: The design is carefully made, although the major flaw - not mentioned by the authors as a weakness - is it depends on self-action by the patients and involves subjective measurement as pain. It should be clearly explained and stated in discussion as it is a has a major influence on the validity and impact of the results.
Response 1: We agree with the reviewer. We added your comments to the discussion (line:304-306).
Point 2: The results section should be slightly re-written - the presentation is not always clear and easy for understanding.
Response 2: We agree with the reviewer. We corrected Table1-3 and Figures1,2
Point 3: Finally, would appreciate clearer conclusions with advice for future research.
Response 3: We agree with the reviewer. We corrected the conclusions to make them clearer.